# Differential bioenergetic profile of human glioblastoma following transplantation of myocyte-derived mitochondria

Kent L. Marshall[1], Ethan Meadows[2,3], Alan Mizener[4], John M. Hollander[2,3], Christopher P. Cifarelli[1,5*]

1 Department of Neurosurgery, West Virginia University, Morgantown, West Virginia, United States of America, 2 Department of Physiology, Pharmacology & Toxicology, West Virginia University, Morgantown, West Virginia, United States of America, 3 Mitochondria, Metabolism & Bioenergetics Working Group, West Virginia University, Morgantown, West Virginia, United States of America, 4 Mary Babb Randolph Cancer Institute, West Virginia University, Morgantown, West Virginia, United States of America, 5 Department of Radiation Oncology, West Virginia University, Morgantown, West Virginia, United States of America

* cpcifarelli@hsc.wvu.edu

## Abstract

Glioblastoma (GBM) exhibits profound plasticity, enabling adaptation to fluctuating microenvironmental stressors such as hypoxia and nutrient deprivation. However, this metabolic rewiring also creates subtype-specific vulnerabilities that may be exploited therapeutically. Here, we investigate whether mitochondrial transplantation using non-neoplastic, human myocyte-derived mitochondria alters the metabolic architecture of GBM cells and modulates their response to ionizing radiation. Using a cell-penetrating peptide–mediated delivery system, we successfully introduced mitochondria into two mesenchymal-subtype GBM cell lines, U3035 and U3046. Transplanted cells exhibited enhanced mitochondrial polarization and respiratory function, particularly in the metabolically flexible U3035 line. Bioenergetic profiling revealed significant increases in basal respiration, spare respiratory capacity, and glycolytic reserve in U3035 cells post-transplantation, whereas U3046 cells showed minimal bioenergetic augmentation. Transcriptomic analyses using oxidative phosphorylation (OXPHOS) and glycolysis gene sets confirmed these functional findings. At baseline, U3035 cells expressed high levels of both glycolytic and OXPHOS genes, while U3046 cells were metabolically suppressed. Following radiation, U3035 cells downregulated key OXPHOS and glycolysis genes, suggesting metabolic collapse. In contrast, U3046 cells transcriptionally upregulated both pathways, indicating compensatory adaptation. These results identify and establish mitochondrial transplantation as a metabolic priming strategy that sensitizes adaptable GBM subtypes like U3035 to therapeutic stress by inducing bioenergetic overextension. Conversely, rigid subtypes like U3046 may require inhibition of post-radiation metabolic compensation for

**Data availability statement:** All relevant data are within the manuscript and its Supporting information files.

**Funding:** NIGMS P20GM121322 and NIH U54GM104942 to CPC. R01 HL-168290 and R01 ES-034628 to JMH. Community Foundation for the Ohio Valley Whipkey Trust to JMH.

**Competing interests:** The authors have declared that no competing interests exist.

effective targeting. Our findings support a novel stratified approach to GBM treatment which integrates metabolic subtype profiling with bioenergetic modulation.

## Introduction

Metabolic reprogramming in glioblastoma (GBM) has traditionally been investigated in the context of cellular adaptation to hypoxic or nutrient-deprived environments [1,2]. However, accumulating evidence suggests that these metabolic shifts are not merely survival mechanisms but are intrinsic to GBM progression and its capacity to remodel and leverage the tumor microenvironment [3]. One hallmark of this adaptability is the ability of GBM cells to shift energy production from oxidative phosphorylation (OXPHOS) to glycolysis—a phenomenon first described by Warburg over a century ago [4,5]. Despite its significant role in tumor metabolism, this metabolic plasticity and the related dynamic flux of mitochondrial density and fucntion remains largely untargeted and underexploited by current therapeutic paradigms.

The efficiency of energy production, particularly through mitochondrial function, underpins the rapid proliferation and invasiveness of GBM cells. Unlike classical mitochondrial disorders such as mitochondrial encephalopathy, lactic acidosis, and stroke-like episodes (MELAS), where deficits in respiration are well-characterized, the disruption of mitochondrial energy coupling in GBM is more nuanced. While functional mitochondrial transplantation has been shown to restore respiratory capacity in cells with inherent mitochondrial dysfunction, its implications in cancer cells—especially those with suppressed OXPHOS activity—remain poorly understood [6].

Recent work by Marshall et al. (2025) demonstrated the feasibility of introducing exogenous GBM-derived mitochondria into genetically distinct recipient GBM cells, with sustained retention and enhanced radiation-induced reactive oxygen species (ROS) production compared to unmodified cells [7]. However, from a translational perspective, practical therapeutic application of this concept would require the use of autologous, non-neoplastic mitochondria to avoid enhancement of malignancy and immunogenicity [8,9]. Therefore, it is imperative to delineate that such mitochondrial transfer either potentiates the efficacy of standard treatments (e.g., radiation or chemotherapy) or independently contributes to a metabolic "de-reprogramming" of GBM cells away from their glycolytic phenotype.

In the present study, we build upon the foundation of mitochondrial transplant into GBM and advance the concept of using non-neoplastic donors into tumor cell recipients by demonstrating successful mitochondrial transplantation into multiple GBM cell lines using human myocyte-derived mitochondria. Utilizing a cell-penetrating peptide (CPP)-mediated delivery system, we specifically assessed the impact of this intervention on both basal and reserve metabolic activity in GBM cells post-transplantation in various states of cellular stress from hypoxia, nutrient deprivation, to radiation exposure.

## Materials and methods

**Cell Lines.** Two human mesenchymal-subtype glioblastoma cell lines, U3035 and U3046, were obtained from the Human Glioblastoma Cell Culture (HGCC) Biobank at Uppsala University (Uppsala, Sweden) [10]. Cells were cultured at 37°C in 5% $CO_2$ using DMEM/F-12 medium supplemented with EGF, FGF, N2, B27, and penicillin/streptomycin, following standard passaging protocols. Primary human skeletal muscle cells (PCS 950−010; ATCC, Manassas, VA) were maintained in mesenchymal stem cell basal medium with L-glutamine, EGF, dexamethasone, FGF-b (5 ng/mL), insulin, and stem cell-qualified FBS. Cell line identity was confirmed by short tandem repeat (STR) analysis (*supplemental data*). A subset of U3035, U3046, and their mitochondrial transplant derivatives were cultured under hypoxia (1% $O_2$) using a Cytocentric® C-Chamber (Biospherix, Parish, NY) to evaluate mitochondrial responses under low oxygen conditions.

**Mitochondrial Isolation and Quantification.** PCS cells were grown in T75 flasks to 90% confluency, dissociated with Accutase® (MilliporeSigma, Burlington, MA), and pelleted by centrifugation. After resuspension in mitochondrial isolation buffer, cell counts were obtained using a Cellometer® Auto 2000 (Nexcelom, Waltham, MA). Mitochondria were extracted using a commercial kit (Abcam, Waltham, MA) via reagent-based permeabilization and differential centrifugation ($600 \times g$ and $7000 \times g$). Isolates were stored in buffer and quantified by optical density (OD A280) using a BioTek Take3 plate reader (Agilent, Santa Clara, CA).

**Mitochondrial Labeling and Transplantation.** Mitochondria and recipient cells were labeled using MitoTracker™ dyes (Thermo Fisher, Waltham, MA). U3035 and U3046 cells were seeded at $5 \times 10^5$ cells/well in 6-well plates and stained with green MitoTracker™ (200 nM) for 30 minutes at 37°C. Donor mitochondria were similarly stained with red MitoTracker™ and incubated with Pep-1-Cysteamine (Pep-1, Anspec, Fremont, CA) cell-penetrating peptide (0.06 mg Pep-1 per 100 µg mitochondrial protein) for 20 minutes at room temperature. The conjugated mitochondria (100 µg) were added to recipient cells and incubated for 48 hours. Transplantation success was imaged using the BioTek™ Cytation 5 system. Individual cells were segmented and analyzed for red/green channel intensities (469/525 nm, 586/647 nm) and visualized using violin plots and XY scatter plots in GraphPad Prism (v10.4.0).

**JC-1 Assay for Mitochondrial Membrane Potential.** Mitochondrial membrane potential (ΔΨm) was assessed in U3035, U3046, and PCS-transplanted variants using JC-1 dye (Thermo Fisher, Waltham, MA). Cells were plated in 12-well plates and cultured under normoxia or hypoxia (1% $O_2$), with high (4 g/L) or low (1 g/L) glucose media. JC-1 (2 µM) was applied in warm complete medium for 30 minutes at 37°C. After PBS wash, cells were imaged using the Cytation 5 platform. Red (aggregated JC-1; 535/590 nm) and green (monomeric JC-1; 485/530 nm) fluorescence were captured, and red/green ratios were calculated per cell. Each condition was tested in duplicate/triplicate across three biological replicates. For oxidative stress studies, cells were exposed to 8 Gy of ionizing radiation (220 kV, 13 mA) using a XenX platform (Xstrahl, Suwanee, GA). Prior dose optimization had identified 8 Gy as effective in U3035 and U3046 lines [7,11].

**Seahorse XF Mito Stress Assay.** Bioenergetic profiling was conducted using the Seahorse XF Cell Mito Stress Test (Agilent, Santa Clara, CA) on XF Pro M plates. U3035 and U3046 cells were seeded at $4 \times 10^4$ cells/well and incubated overnight. Prior to the assay, cells were equilibrated in Seahorse XF Assay Medium (DMEM with 10 mM glucose, 1 mM pyruvate, 2 mM glutamine, pH 7.4) for one hour at 37°C in a non-$CO_2$ incubator. OCR was measured after sequential injections of oligomycin (1.5 µM), FCCP (0.5 µM), and rotenone/antimycin A (0.5 µM each). Basal respiration, ATP-linked respiration, maximal respiration, proton leak, and spare respiratory capacity were calculated. Cells were dissociated post-assay using Accutase® and counted via Cellometer Auto T4. OCR values were normalized to cell number. All conditions were tested in six technical replicates. Data were analyzed using Agilent Wave and GraphPad Prism (v. 10.5.0).

**Transcriptomic Profiling.** U3035 and U3046 cells were cultured under standard conditions, with a subset exposed to 8 Gy radiation (220 kV, 13 mA, 3.53 Gy/min). Six hours post-irradiation, total RNA was extracted using the RNeasy Mini Kit (Qiagen, Santa Clarita, CA). RNA quality was confirmed via Qubit fluorometry and Agilent 2100 Bioanalyzer; only samples with RIN > 7.0 were used. Libraries were prepared using the KAPA mRNA HyperPrep Kit and sequenced on the

Illumina NextSeq 2000 platform (PE50, paired-end). Sequence data were uploaded to NCBI Bioproject (PRJNA1288533) and differential expression analysis was conducted following transcript quantification. Expression heatmaps were generated using $\log_2$-transformed, normalized values for KEGG oxidative phosphorylation pathway genes (KEGG ID: 33467). Hierarchical clustering (Euclidean distance) was used to visualize gene expression shifts by cell line and treatment condition. For pathway-level analysis, $\log_2FC$ values were mapped onto KEGG metabolic diagrams using Pathview R package (v1.38.0), enabling visual integration of transcriptomic data within curated pathway structures [12,13]. Genes were color-coded based on $\log_2FC$ magnitude to reflect subtype-specific transcriptional responses to radiation.

## Results

### Feasibility of myocyte-derived mitochondrial transplantation into GBM cells

To evaluate the feasibility of delivering exogenous mitochondria into glioblastoma (GBM) cells, mitochondria were isolated from skeletal myocytes, labeled with MitoTracker™ Red, and conjugated with the cell-penetrating peptide Pep-1 to facilitate cellular entry. U3035 and U3046 GBM cells were incubated with the Pep-1–mitochondria complex, and fluorescence analysis was performed 48 hours post-transplantation. As a reference for baseline endogenous mitochondrial content, U3035 and U3046 cells were stained with MitoTracker™ Green prior to mitochondrial transplant (Fig 1B, 1H). In U3035 cells, the mean fluorescence intensity (MFI) for native mitochondria was 5,310, while U3046 cells exhibited an MFI of 5,848. In contrast, cells exposed to myocyte-derived mitochondria-Pep-1 conjugates stained with MitoTracker™ Red displayed MFIs of 11,980 (U3035) and 16,780 (U3046) (Fig 1E, 1F, 1K, 1L). These data confirm successful delivery of exogenous mitochondria into both GBM cell lines and demonstrate a higher MitoTracker™ Red signal in U3046 relative to U3035 (Fig 1C, 1I).

### Mitochondrial polarization is enhanced in transplanted cells under basal and post-radiation conditions

Mitochondrial membrane potential (ΔΨm), assessed by JC-1 red-to-green fluorescence ratio, revealed enhanced mitochondrial polarization in mitochondria-transplanted (PCS) GBM cells compared to their wild-type (MT) counterparts under both normoxic and radiation stress conditions. Under basal conditions, U3046 PCS cells consistently demonstrated higher mean JC-1 ratios across all glucose and oxygen conditions, including normoxic high glucose (3.18 vs. 2.17 in MT), normoxic low glucose (2.86 vs. 1.39), and hypoxic low glucose (2.75 vs. 2.66) (Fig 2A, B). U3035 PCS cells also exhibited elevated JC-1 ratios relative to MT cells, most notably under hypoxic low glucose (5.21 vs. 2.71), suggesting a stronger polarization phenotype particularly under nutrient stress (Fig 2A, B).

Following exposure to 8 Gy ionizing radiation, JC-1 ratios declined across all conditions, consistent with radiation-induced mitochondrial depolarization. However, transplanted PCS cells retained higher ΔΨm relative to MT cells. In U3046 cells, the JC-1 ratio remained elevated in PCS vs. MT groups under both normoxic high glucose (0.67 vs. 0.48) and hypoxic low glucose (1.75 vs. 0.63). Similarly, U3035 PCS cells showed higher JC-1 values than MT cells under normoxic high glucose (0.49 vs. 0.64) and hypoxic high glucose (1.78 vs. 1.33), though the magnitude of difference varied by condition (Fig 2C, D).

**Transplanted mitochondria augment bioenergetic capacity in metabolically flexible GBM cells.** Seahorse XF Mito Stress Tests revealed that mitochondrial transplantation significantly enhanced oxidative phosphorylation in the metabolically flexible U3035 glioblastoma (GBM) cell line. Compared to wild-type controls (MT), U3035 PCS cells exhibited a 1.7-fold increase in basal respiration (p = 0.003), a 2.1-fold increase in maximal respiration following FCCP treatment (p = 0.001), and a 2.5-fold elevation in spare respiratory capacity (SRC) (p < 0.001). These changes indicate robust bioenergetic enhancement following exogenous mitochondrial integration. In addition, PCS cells displayed a significantly higher extracellular acidification rate (ECAR) post-oligomycin injection (p = 0.012), reflecting increased glycolytic reserve and metabolic plasticity (Fig 3A, 3D).

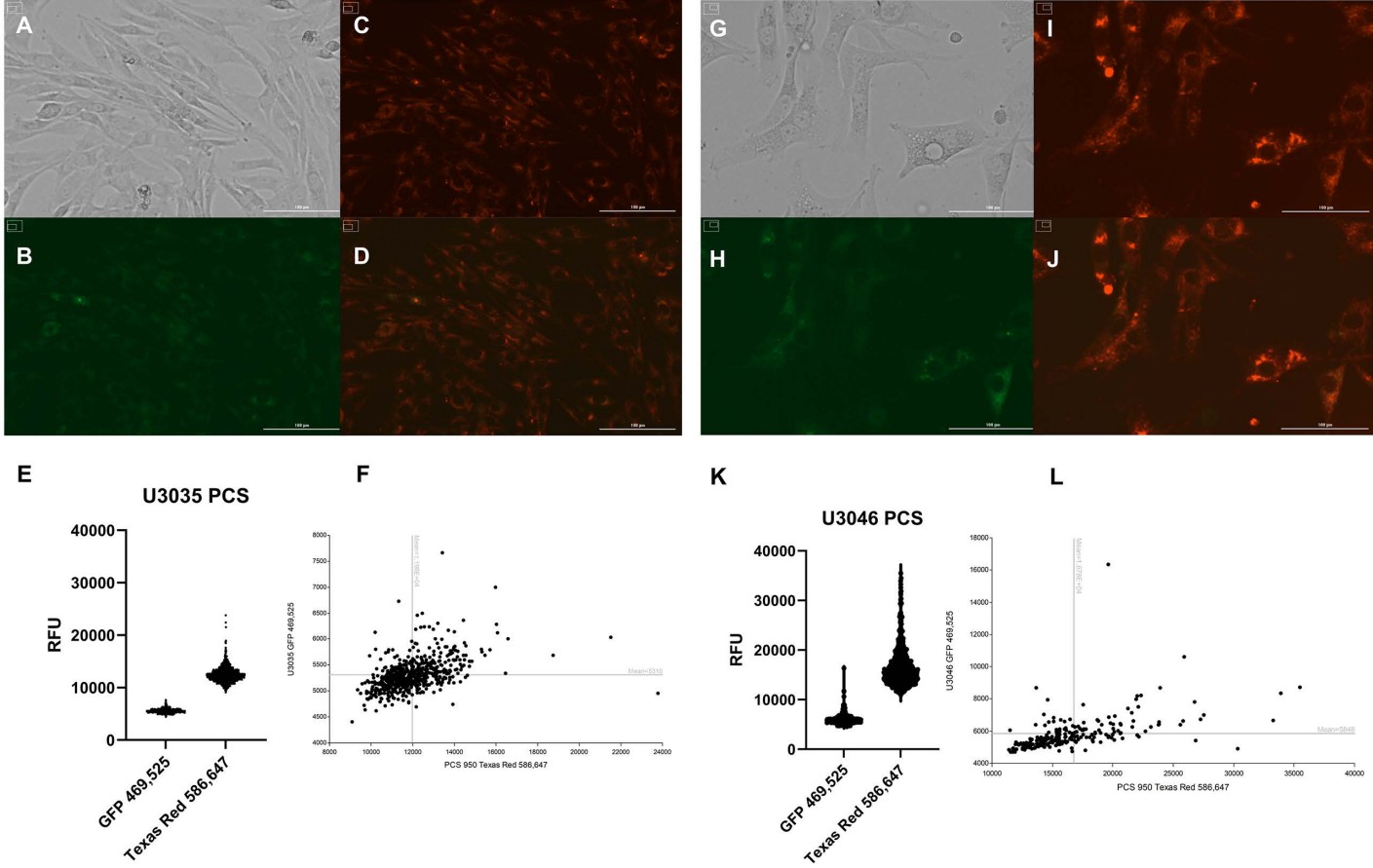

**Fig 1. Quantification of mitochondrial uptake following Pep-1–mediated transplantation into GBM cells.** U3035 and U3046 glioblastoma (GBM) cells were stained with MitoTracker™ Green to label endogenous mitochondria (A & H) and incubated with MitoTracker™ Red–labeled myocyte-derived mitochondria conjugated with the Pep-1 cell-penetrating peptide (C & I). Mean fluorescence intensity (MFI) was measured 48 hours post-transplantation using Cytation 5 imaging. In U3035 cells, MFI values were 5,310 (green) and 11,980 (red), whereas U3046 cells showed 5,848 (green) and 16,780 (red), respectively (E, F, K, L). Corresponding brightfield and overlay images confirm co-localization of native and exogenous mitochondria (A, D, G, J). Each bar represents the mean of >500 segmented cells per condition from at least two independent experiments.

In contrast, U3046 PCS cells showed only minor increases in mitochondrial respiration. Basal OCR was not significantly different from MT (p = 0.41), and while maximal respiration and SRC showed slight increases (1.2-fold and 1.3-fold, respectively), these changes did not reach statistical significance (p = 0.18 and p = 0.22, respectively). ECAR measurements also showed no significant difference between U3046 PCS and MT cells (p = 0.47), indicating limited glycolytic responsiveness (Fig 3B, 3E).

## Transcriptomic profiling reveals distinct metabolic signatures and compensatory pathways between GBM subtypes

RNA seqeunce data were uploaded to NCBI Bioproject under the accession number PRJNA128853. Transcriptomic analysis of U3035 and U3046 glioblastoma (GBM) cells under basal and post-radiation conditions revealed subtype-specific differences in metabolic gene regulation involving both oxidative phosphorylation (OXPHOS) and glycolysis pathways. Using the KEGG Oxidative Phosphorylation gene set (133 genes), 111 were detected in the RNA-seq dataset. Hierarchical clustering and heatmap analysis clearly separated U3035 and U3046 cells by both cell type and treatment status.

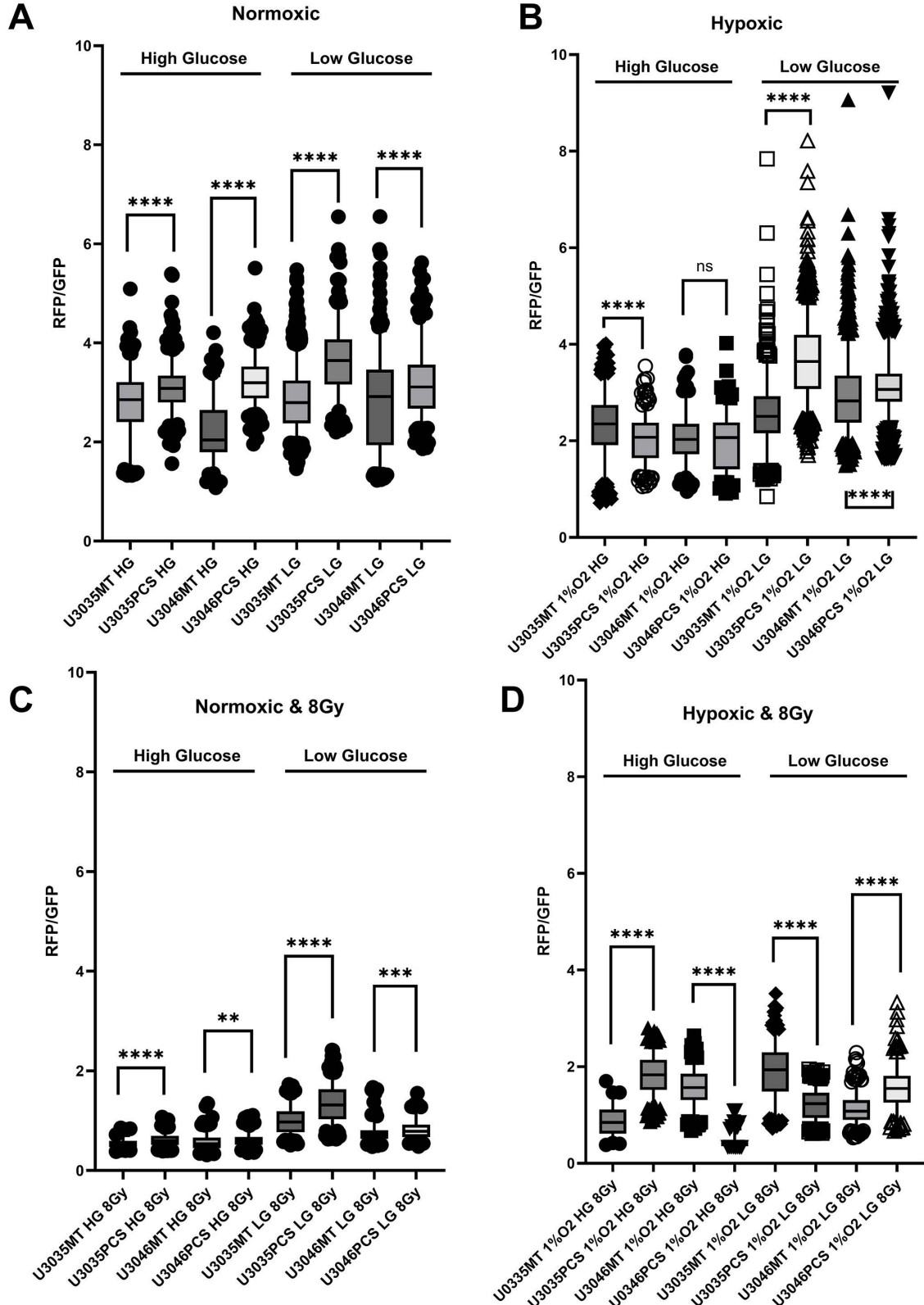

**Fig 2. Mitochondrial polarization is enhanced by mitochondrial transplantation in GBM cells across metabolic and radiation stress conditions.** JC-1 red:green fluorescence ratios were measured at the single-cell level in U3035 and U3046 glioblastoma cells 48 hours after myocyte-derived

mitochondrial transplantation (PCS) or control treatment (MT). Cells were cultured under combinations of oxygen (normoxia (A & C) vs. hypoxia; (B & D)), glucose (low vs. high), and radiation (0 Gy (A & B) vs. 8 Gy (C & D)) conditions to simulate metabolic stress. Across all tested environments, mitochondrial transplantation significantly increased mitochondrial membrane potential (ΔΨm), as evidenced by elevated JC-1 red:green fluorescence ratios. Under normoxic, high-glucose conditions without radiation, mean JC-1 ratios increased from 2.80 to 3.09 in U3035 cells (n = 298 and 430, respectively) and from 2.21 to 3.23 in U3046 cells (n = 170 and 348), with p < 0.0001 in both comparisons. Following 8 Gy radiation exposure, U3035 PCS cells maintained significantly elevated polarization (MT: 0.531 vs. PCS: 0.709, n = 456 each, p < 0.0001), and U3046 PCS cells also showed an increase (MT: 0.390 vs. PCS: 0.467, n = 456 each, p = 0.0046).

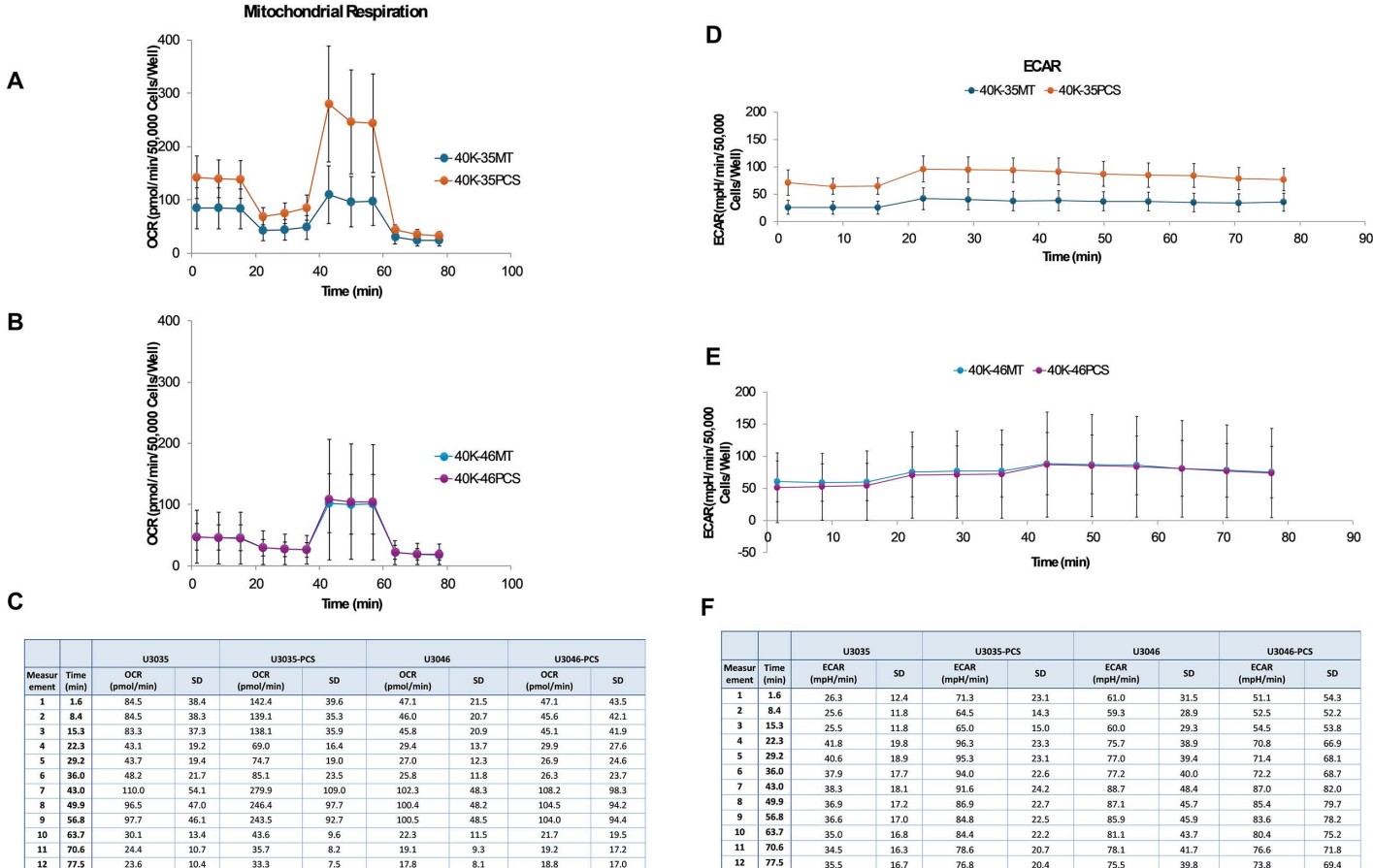

**Table C. U3035 / U3035-PCS / U3046 / U3046-PCS OCR data**

| Measurement | Time (min) | U3035 OCR (pmol/min) | SD | U3035-PCS OCR (pmol/min) | SD | U3046 OCR (pmol/min) | SD | U3046-PCS OCR (pmol/min) | SD |
|---|---|---|---|---|---|---|---|---|---|
| 1 | 1.6 | 84.5 | 38.4 | 142.4 | 39.6 | 47.1 | 21.5 | 47.1 | 43.5 |
| 2 | 8.4 | 84.5 | 38.3 | 139.1 | 35.3 | 46.0 | 20.7 | 45.6 | 42.1 |
| 3 | 15.3 | 83.3 | 37.3 | 138.1 | 35.9 | 45.8 | 20.9 | 45.1 | 41.9 |
| 4 | 22.3 | 43.1 | 19.2 | 69.0 | 16.4 | 29.4 | 13.7 | 29.9 | 27.6 |
| 5 | 29.2 | 43.7 | 19.4 | 74.7 | 19.0 | 27.0 | 12.3 | 26.9 | 24.6 |
| 6 | 36.0 | 48.2 | 21.7 | 85.1 | 23.5 | 25.8 | 11.8 | 26.3 | 23.7 |
| 7 | 43.0 | 110.0 | 54.1 | 279.9 | 109.0 | 102.3 | 48.3 | 108.2 | 98.3 |
| 8 | 49.9 | 96.5 | 47.0 | 246.4 | 97.7 | 100.4 | 48.2 | 104.5 | 94.2 |
| 9 | 56.8 | 97.7 | 46.1 | 243.5 | 92.7 | 100.5 | 48.5 | 104.0 | 94.4 |
| 10 | 63.7 | 30.1 | 13.4 | 43.6 | 9.6 | 22.3 | 11.5 | 21.7 | 19.5 |
| 11 | 70.6 | 24.4 | 10.7 | 35.7 | 8.2 | 19.1 | 9.3 | 19.2 | 17.2 |
| 12 | 77.5 | 23.6 | 10.4 | 33.3 | 7.5 | 17.8 | 8.1 | 18.8 | 17.0 |

**Table F. U3035 / U3035-PCS / U3046 / U3046-PCS ECAR data**

| Measurement | Time (min) | U3035 ECAR (mpH/min) | SD | U3035-PCS ECAR (mpH/min) | SD | U3046 ECAR (mpH/min) | SD | U3046-PCS ECAR (mpH/min) | SD |
|---|---|---|---|---|---|---|---|---|---|
| 1 | 1.6 | 26.3 | 12.4 | 71.3 | 23.1 | 61.0 | 31.5 | 51.1 | 54.3 |
| 2 | 8.4 | 25.6 | 11.8 | 64.5 | 14.3 | 59.3 | 28.9 | 52.5 | 52.2 |
| 3 | 15.3 | 25.5 | 11.8 | 65.0 | 15.0 | 60.0 | 29.3 | 54.5 | 53.8 |
| 4 | 22.3 | 41.8 | 19.8 | 96.3 | 23.3 | 75.7 | 38.9 | 70.8 | 66.9 |
| 5 | 29.2 | 40.6 | 18.9 | 95.3 | 23.1 | 77.0 | 39.4 | 71.4 | 68.1 |
| 6 | 36.0 | 37.9 | 17.7 | 94.0 | 22.6 | 77.2 | 40.0 | 72.2 | 68.7 |
| 7 | 43.0 | 38.3 | 18.1 | 91.6 | 24.2 | 88.7 | 48.4 | 87.0 | 82.0 |
| 8 | 49.9 | 36.9 | 17.2 | 86.9 | 22.7 | 87.1 | 45.7 | 85.4 | 79.7 |
| 9 | 56.8 | 36.6 | 17.0 | 84.8 | 22.5 | 85.9 | 45.9 | 83.6 | 78.2 |
| 10 | 63.7 | 35.0 | 16.8 | 84.4 | 22.2 | 81.1 | 43.7 | 80.4 | 75.2 |
| 11 | 70.6 | 34.5 | 16.3 | 78.6 | 20.7 | 78.1 | 41.7 | 76.6 | 71.8 |
| 12 | 77.5 | 35.5 | 16.7 | 76.8 | 20.4 | 75.5 | 39.8 | 73.8 | 69.4 |

**Fig 3. Bioenergetic Capacity in Metabolically Flexible GBM Cells.** Seahorse XF Mito Stress Test analysis of oxygen consumption rate (OCR) and extracellular acidification rate (ECAR) in U3035 and U3046 glioblastoma (GBM) cells following transplantation with myocyte-derived mitochondria. (A & B) OCR over time in U3035 cells (40K-35MT: non-transplanted; 40K-35PCS: post-transplant) (C) Mean OCR data at each time point with standard deviation (SD). ECAR over time in U3035 and U3046 cells comparing MT and PCS conditions. (D &E). (F) Mean ECAR data at each time point with standard deviation (SD). Data are normalized per 50,000 cells per well. Each condition was tested using six technical replicates. Traces represent the mean response across six technical replicates for each condition.

At baseline, U3035 cells demonstrated broad upregulation of OXPHOS-related transcripts, including ATP5O, COX7C, and NDUFS3, indicating a robust mitochondrial bioenergetic state. In contrast, U3046 cells showed widespread suppression of mitochondrial gene expression, particularly within Complex I and IV components such as NDUFA9, COX6A2, and NDUFB6, consistent with a metabolically less oxidative and more rigid phenotype (Fig 4A).

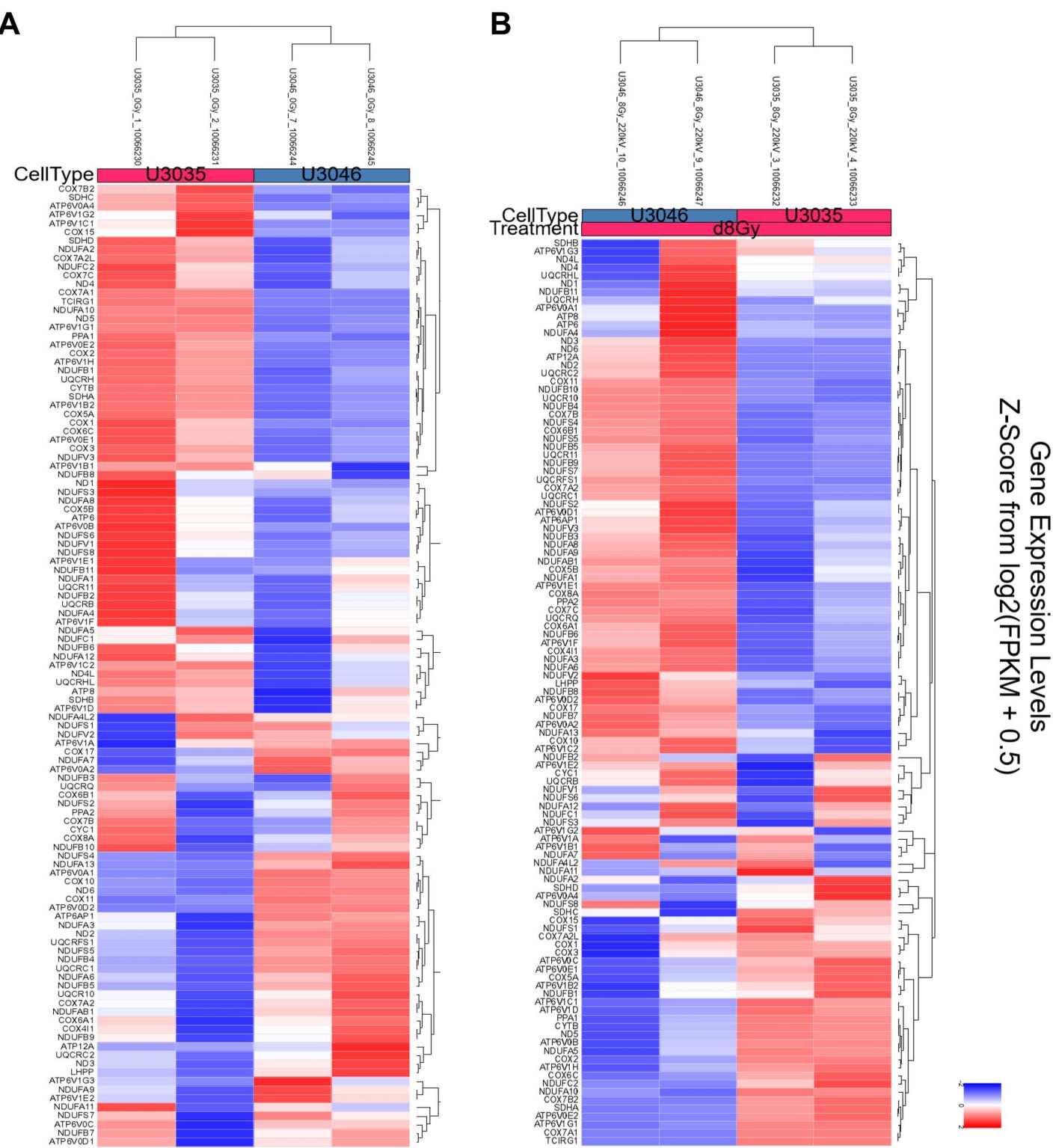

**Fig 4. Transcriptomic heatmaps illustrating differential expression of oxidative phosphorylation (OXPHOS) pathway genes in U3035 and U3046 glioblastoma cells under basal and post-radiation conditions.** (A) Heatmap of $\log_2$-transformed expression values from 111 detected genes within the KEGG_OXPHOS pathway (out of 133 total) under non-irradiated (0Gy) conditions. (B) Heatmap of the same gene set following 8 Gy ionizing radiation. Hierarchical clustering demonstrated distinct subtype-specific expression patterns at baseline and post-treatment.

Following 8 Gy radiation, the two cell lines diverged in their mitochondrial stress responses. U3046 cells exhibited increased expression of several OXPHOS genes—NDUFB4, UQCR10, NDUFB10, COX7B, and NDUFS4 (log$_2$FC ~ 1.98–1.99)—suggesting a coordinated compensatory up-regulation of mitochondrial function. Conversely, U3035 cells selectively down-regulated critical OXPHOS components including TCIRG1, COX7A1, ATP6V1G1, NDUFA10, and COX7B2 (log$_2$FC ~ –1.98 to –2.00), suggesting radiation-induced mitochondrial repression or metabolic destabilization (Fig 4B).

Complementary analysis of glycolytic regulation using the HALLMARK_GLYCOLYSIS functional gene set (197 genes total) identified 188 genes within the dataset. Basally, U3035 cells expressed significantly higher levels of glycolysis-associated transcripts, including ENO2, PGK1, GPI, HK2, and LDHA, indicative of metabolic flexibility and dual-pathway engagement (glycolysis + OXPHOS). U3046 cells displayed comparatively lower expression of key glycolytic enzymes, consistent with a less adaptable metabolic phenotype (Fig 5).

Following radiation exposure, U3035 cells exhibited a modest down-regulation in select glycolytic genes such as ALDOC, PFKL, and PGAM1, though overall glycolytic output appeared sustained. In contrast, U3046 cells initiated a pronounced up-regulation of glycolysis-related genes including GPI, HK2, ENO1, and LDHA, reflecting a radiation-induced shift toward glycolytic compensation (Fig 5).

## Pathway-level transcriptomic analysis highlights divergent metabolic adaptation following radiation exposure

To assess broader functional responses to radiation at the pathway level, KEGG pathway enrichment analysis was performed on the oxidative phosphorylation (hsa00190) and glycolysis/gluconeogenesis (hsa00010) gene sets using log$_2$ fold-change (log$_2$FC) values for U3035 and U3046 cells. These data extend single-gene analyses by mapping radiation-induced transcriptional dynamics across key energy metabolism modules.

For the oxidative phosphorylation pathway (133 genes), radiation induced distinct responses in the two GBM cell lines. In U3035 cells, a net down-regulation of OXPHOS components was observed, particularly among genes encoding Complex I and Complex IV subunits, including NDUFA10, COX7B2, and ATP6V1G1. In contrast, U3046 cells exhibited coordinated up-regulation of several mitochondrial genes post-radiation, including NDUFB4, UQCR10, and COX7B, consistent with a compensatory re-engagement of mitochondrial respiration under genotoxic stress. These pathway-wide shifts mirror previously noted single-gene trends (Fig 6A).

Analysis of the glycolysis/gluconeogenesis gene set (197 genes total; 188 detected) similarly revealed subtype-specific divergence. U3035 cells, which exhibited high basal glycolytic expression (e.g., ENO2, HK2, LDHA), showed modest down-regulation of glycolytic enzymes post-radiation, including ALDOC, PFKL, and PGAM1, but maintained overall pathway activity. In contrast, U3046 cells up-regulated multiple glycolysis-related genes in response to radiation, including ENO1, GPI, HK2, and LDHA (Fig 6B).

## Discussion

Metabolic adaptability is increasingly recognized as a defining feature of glioblastoma (GBM), enabling dynamic resistance responses to therapeutic stress and facilitating sustained tumor progression [14,15]. Our study demonstrates that this metabolic adaptability can be leveraged through mitochondrial transplantation. It also demonstrates that mitochondrial transplantation elicits a granular, subtype-specific metabolic outcomes in different GBM cells, shaped by intrinsic differences in metabolic architecture and stress-response programming. U3035 and U3046 cells exhibited marked divergence in mitochondrial polarization, bioenergetic profiles, and transcriptional adaptation to radiation-induced stress. These phenotypes are consistent with previous observations that Myc-amplified GBM subtypes—typified by elevated glycolytic flux—exhibit heightened sensitivity to metabolic perturbation, particularly via NAD$^+$ salvage inhibition [16].

Flexibility in metabolic activity across GBM as a clinical entity is more nuanced at the individual level as well. Recent work has redefined GBM as a disease with distinct metabolic subtypes characterized by transcriptional and mitochondrial features that confer therapeutic vulnerabilities. Garofano et al [17] identified a mitochondrial GBM subtype defined

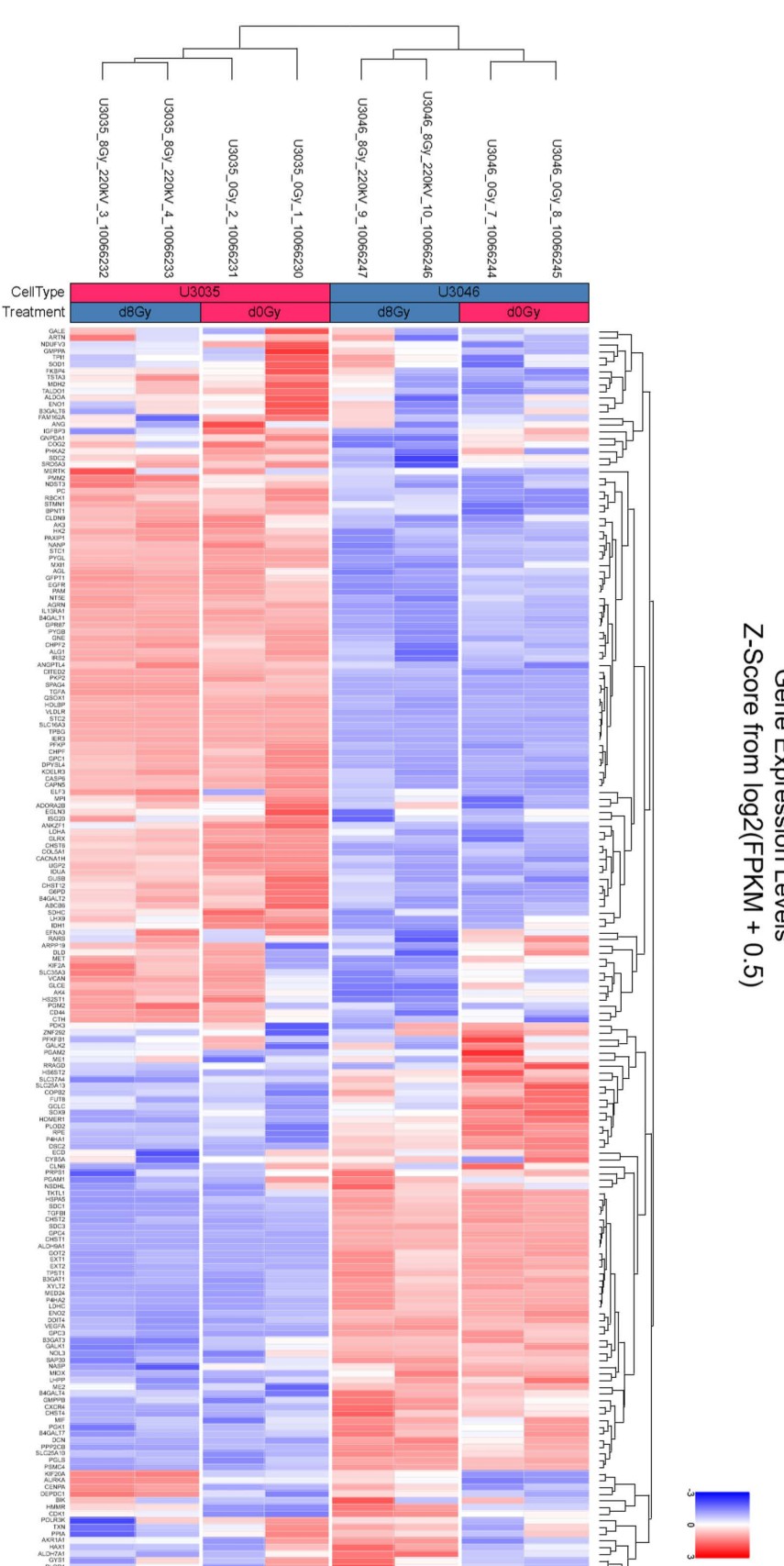

**Fig 5. Heatmap depicting expression of glycolysis-associated genes in U3035 and U3046 glioblastoma cells under basal and post-radiation (8 Gy) conditions.** Expression levels represent log₂-transformed values for 188 detected transcripts from the HALLMARK_GLYCOLYSIS gene set. Hierarchical clustering was applied across genes and treatment groups, illustrating differential baseline expression and radiation-induced transcriptional shifts between GBM subtypes.

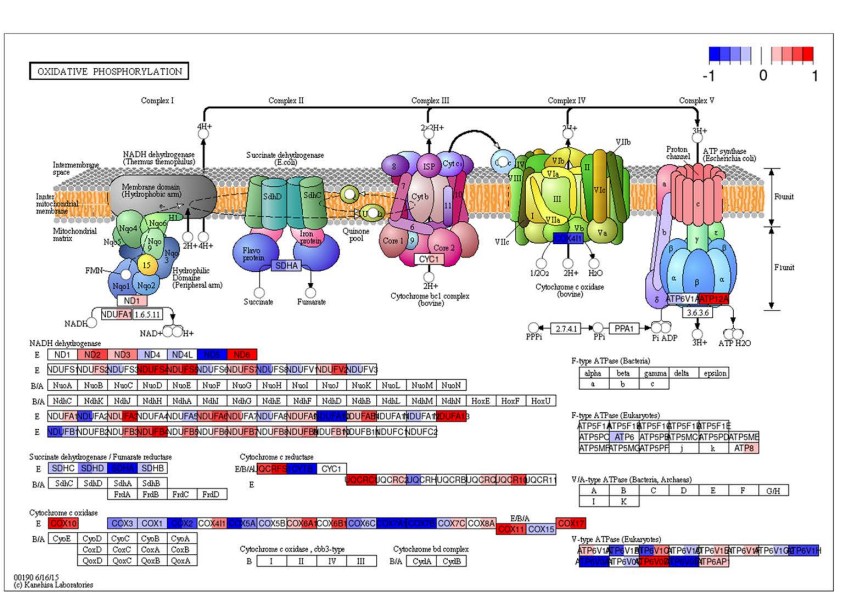

**A**

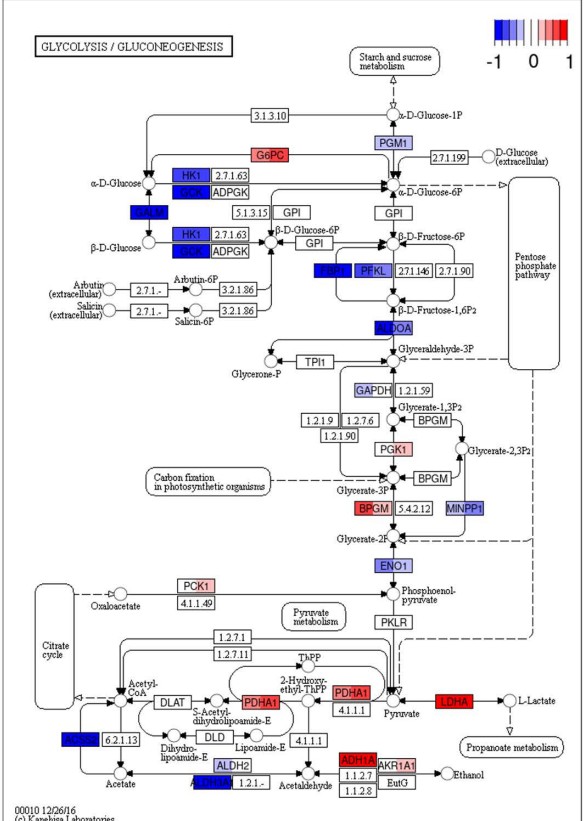

**B**

**Fig 6. Transcriptomic comparison of oxidative phosphorylation (OxPhos) gene expression in U3046 vs. U3035 GBM cell lines.** (A) KEGG pathway map of oxidative phosphorylation (hsa00190) displaying differential gene expression between U3046 (left half of each color bar) and U3035 (right half of each color bar) under basal conditions. Heatmap coloring reflects log₂ fold-change (log₂FC) values of irradiated cells relative to baseline, un-irradiated gene expression, with the color scale ranging from −1.0 (blue, down-regulated) to +1.0 (red, up-regulated), as indicated by the legend bar (top right). Notably, U3035 cells exhibited broad up-regulation of Complex I (NDUFS, NDUFA) and Complex V (ATP5, ATP6) subunits, consistent with enhanced mitochondrial bioenergetic capacity. In contrast, U3046 cells showed pronounced repression across several complexes, including down-regulation of Complex I and IV components (e.g., NDUFB8, COX11), indicative of a less active OXPHOS phenotype. Select Complex II components (e.g., SDHA, SDHB) were modestly preserved in both lines. (B) KEGG pathway map of glycolysis/gluconeogenesis (hsa00010) displaying differential gene expression between U3046 (left half of each color bar) and U3035 (right half of each color bar) under basal (0 Gy) conditions. U3035 cells demonstrated up-regulation of glycolytic enzymes involved in glucose phosphorylation and pyruvate metabolism (e.g., HK1, PFKL, ENO1, LDHA), consistent with enhanced glycolytic engagement. In contrast, U3046 cells showed widespread repression of early and intermediate glycolytic enzymes (e.g., GALM, GAPDH, ALDH3A1), with relative up-regulation of select gluconeogenic and pyruvate metabolism genes (e.g., PCK1, PDHA1), suggestive of distinct carbon flux regulation. KEGG pathway components reprinted from Kanehisa et al [12] under a CC BY license, with permission from Oxford Academic original copyright 2023.

by enhanced OXPHOS activity, which was selectively sensitive to metabolic perturbation and redox imbalance. These findings parallel our observations that mitochondrial transplantation augments redox resilience, particularly in U3035 cells, by stabilizing mitochondrial membrane potential (ΔΨm) and supporting oxidative capacity under genotoxic stress. Using a Pep-1 peptide–mediated delivery platform, we achieved efficient mitochondrial transfer from skeletal myocytes, confirmed by fluorescent signal tracking. CPPs, such as Pep-1 have a substantial amount of data supporting the efficiency of transport into a donor cell [18,19]. Moreover, recent studies have confirmed that this approach can be clinically translatable with no evidence of toxicity [20]. Notably, U3046 cells exhibited more persistent mitochondrial signal post-transplantation, possibly reflecting enhanced uptake or reduced turnover. Our work also indicates that transplanted mitochondria maintain polarization even under nutrient-limited or hypoxic conditions and reduced radiation-induced depolarization, suggesting a stabilizing effect on mitochondrial integrity under metabolic stress.

Functionally, mitochondrial transplantation significantly enhanced both OXPHOS and glycolytic flux in U3035 cells, as revealed by Seahorse assays. In contrast, U3046 cells exhibited metabolic inertia, showing limited improvement in ATP-generating pathways. These findings suggest that mitochondrial augmentation preferentially benefits metabolically plastic cells, with pre-existing OXPHOS/glycolysis engagement (as in U3035) serving as a predictor of responsiveness to therapy. Transcriptomic data reinforced this distinction: U3035 cells expressed higher levels of OXPHOS genes (e.g., ATP5O, COX7C, NDUFS3) and glycolytic enzymes (ENO2, PGK1, HK2, LDHA), indicating a flexible dual-fuel phenotype. U3046 cells, by contrast, displayed baseline repression across both metabolic axes.

Pathway-level transcriptomic analysis further contextualized these findings. Using KEGG-based maps of OXPHOS and glycolysis/gluconeogenesis (hsa00190 and hsa00010), we visualized $\log_2 FC$ shifts following radiation. In U3035 cells, radiation induced coordinated down-regulation of multiple Complex I and IV components (e.g., NDUFA10, NDUFS4, COX7B2) alongside modest suppression of glycolytic regulators (ALDOC, PGAM1), suggesting radiation-induced energetic destabilization. U3046 cells exhibited an inverse pattern, with post-radiation up-regulation of glycolytic effectors (HK2, LDHA, GPI) and increased expression of OXPHOS genes (NDUFB4, UQCR10, COX7B), consistent with a delayed compensatory response. These global pathway shifts mirror the heatmap clustering and reinforce the subtype-specific nature of metabolic stress responses.

Paradoxically, our data suggest that metabolically active subtypes like U3035 are more susceptible to collapse under conditions of redox and energy imbalance, particularly following interventions that perturb mitochondrial integrity, such as radiotherapy. Mitochondrial transplantation may serve not only to bolster energetic capacity but can "prime" adaptable GBM cells for synthetic lethality when combined with redox-disrupting agents such as CPI-613 (PDH inhibitor), rotenone (Complex I inhibitor), or buthionine sulfoximine (GSH synthesis inhibitor) [21,22]. Conversely, more rigid subtypes such as U3046 may require blockade of their delayed compensatory responses, potentially via dual inhibition of glycolysis and ETC flux or NAD$^+$ regeneration pathways [23].

While promising, this study has important limitations. Chief among them is the absence of *in vivo* outcome data and the lack of assessment regarding how mitochondrial transplantation may influence the long-term proliferative capacity of GBM cells. Addressing the former would require rodent model testing, which is complicated by the challenge of selecting a representative cell line—especially given the significant functional differences we observed between just two human GBM lines. Moreover, the increasing recognition of immune system involvement in GBM response would necessitate either the use of rodent glioma lines or humanized models, both of which may limit translatability. Regarding the potential impact on proliferation, our prior study using neoplastic donor and recipient cells did not demonstrate any increase in proliferative index [7]. However, future investigations—particularly those involving different donor:recipient combinations—will need to closely evaluate this possibility.

In sum, our results confirm the technical feasibility and mechanistic relevance of mitochondrial transplantation in GBM. Rather than a direct cytotoxic strategy, mitochondrial augmentation may represent a precision therapeutic approach to unmask latent metabolic liabilities in adaptable GBM subtypes. Stratification based on baseline metabolic engagement

and radiation transcriptional profiles may help identify patients most likely to benefit from metabolism- and redox-directed combination strategies. This builds on our prior work that significant heterogeneity in response exists among glioblastoma cell lines, even those that share mesenchymal gene expression profiles [24]. Further *in vivo* studies will aid in elucidating the impact that mitochondrial transplantation could have in optimizing standard of care treatment for glioblastoma as a therapeutic adjunct.

## Supporting information

**S1 File. Short tandem repeat (STR) analysis was performed to confirm the identity of the U3035 and U3046 cells lines, as well as the corresponding myocyte transplanted U3035 and U3046 cell lines.** All data confirm the correct parental lineage.
(PDF)

## Author contributions

**Conceptualization:** Kent L. Marshall, John M. Hollander, Christopher P. Cifarelli.

**Data curation:** Kent L. Marshall, Ethan Meadows, Alan Mizener, Christopher P. Cifarelli.

**Formal analysis:** Kent L. Marshall, Ethan Meadows, Alan Mizener, John M. Hollander, Christopher P. Cifarelli.

**Funding acquisition:** Christopher P. Cifarelli.

**Investigation:** Kent L. Marshall, Christopher P. Cifarelli.

**Methodology:** Kent L. Marshall, Ethan Meadows, Alan Mizener, John M. Hollander, Christopher P. Cifarelli.

**Supervision:** Christopher P. Cifarelli.

**Validation:** Kent L. Marshall, Alan Mizener, John M. Hollander, Christopher P. Cifarelli.

**Visualization:** Christopher P. Cifarelli.

**Writing – original draft:** Kent L. Marshall, Ethan Meadows, Alan Mizener, John M. Hollander, Christopher P. Cifarelli.

**Writing – review & editing:** Kent L. Marshall, Ethan Meadows, Alan Mizener, John M. Hollander, Christopher P. Cifarelli.

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
