## [Decision Letter · Decision Letter 0]

25 Aug 2025

Dear Dr. Cifarelli,

Thank you for submitting your manuscript to PLOS ONE. After careful consideration, we feel that it has merit but does not fully meet PLOS ONE’s publication criteria as it currently stands. Therefore, we invite you to submit a revised version of the manuscript that addresses the points raised during the review process.

Please respond to reviewers' comments individually.

We look forward to receiving your revised manuscript.

Kind regards,

Xiaosheng Tan

Academic Editor

PLOS ONE

Journal Requirements:

“NIGMS P20GM121322 and NIH U54GM104942 to CPC. 

R01 HL-168290 and R01 ES-034628 to JMH.

Community Foundation for the Ohio Valley Whipkey Trust to JMH”

3. We note that “STR_3035_3046_PCS.pdf” in your submission contain copyrighted images. All PLOS content is published under the Creative Commons Attribution License (CC BY 4.0), which means that the manuscript, images, and Supporting Information files will be freely available online, and any third party is permitted to access, download, copy, distribute, and use these materials in any way, even commercially, with proper attribution. For more information, see our copyright guidelines: http://journals.plos.org/plosone/s/licenses-and-copyright.

1. You may seek permission from the original copyright holder of “STR_3035_3046_PCS.pdf” to publish the content specifically under the CC BY 4.0 license.

Reviewers' comments:

Reviewer's Responses to Questions

**Comments to the Author**

1. Is the manuscript technically sound, and do the data support the conclusions?

Reviewer #1: Yes

Reviewer #2: Yes

2. Has the statistical analysis been performed appropriately and rigorously?

Reviewer #1: Yes

Reviewer #2: Yes

3. Have the authors made all data underlying the findings in their manuscript fully available?

Reviewer #1: Yes

Reviewer #2: Yes

4. Is the manuscript presented in an intelligible fashion and written in standard English?

Reviewer #1: Yes

Reviewer #2: Yes

Reviewer #1: 1.The resolution of all figures need to be improved.

2. The signal of Figure 1-B is weak, seems like background, please improve the quality of picture.

3. For Figure 3A, please show the conducting statistics.

Reviewer #2: The well written and coherent work describes research on mitochondrial transplantation using non-neoplastic, human myocyte-derived mitochondria into two mesenchymal subtype glioblastoma cell lines. The study investigates the impact on metabolic architecture and response to ionizing radiation, combining functional bioenergetic assays and transcriptomic profiling. The work is well presented, conceptualized and scientifically sound. I have several points that could help to improve the already great study.

Questions:

Can the authors comment on the purity and integrity of the isolated mitochondria after transplantation?

A possible limitation is the short post-translation effect (48 hours), could long-term studies improve the understanding of therapeutic durability? In addition, the observation that U3046 cells show higher mitochondrial uptake but less functional augmentation raises questions about mitochondrial turnover, integration, or degradation in different subtypes.

There is minimal discussion on whether mitochondrial transplantation could inadvertently support tumor growth in some contexts or induce off-target effects.

The use of the cell-penetrating peptide Pep-1 is described, but the efficiency, potential toxicity, and specificity of this delivery mechanism are not deeply discussed.

Clarification on statistical methods and whether all replicates are biological or technical would clarify the validity of the findings.

The authors should address how the findings of the work could translate into in vivo models

Minor comments:

In methods, the composition of mitochondrial isolation buffer should be given

Beginning of discussion: “granualr" is mistyped

The picture depicted in fig 6a was created de novo by authors, or adopted from different work? If the latter is true, it should be referenced. The same applies for 6b.

**Do you want your identity to be public for this peer review?** For information about this choice, including consent withdrawal, please see our Privacy Policy

Reviewer #1: No

Reviewer #2: No

---

## [Author Response · Author response to Decision Letter 1]

10 Sep 2025

Responses to Reviewers (PONE-D-25-41544)

Reviewer #1:

1.The resolution of all figures need to be improved.

Thank you for bringing the resolution to our attention. All figures have been formatted according to the publisher’s requirements using TIFF format and 600 dpi.

2. The signal of Figure 1-B is weak, seems like background, please improve the quality of picture.

This figure has been formatted to 600 dpi and the mitochondrial dye is more clearly visible. Once again, this figure represents the baseline mitochondrial content of the cells prior to mitochondrial transplant from the myocyte donor cells, so we anticipated that the intensity of staining would be less than the red tracker dye (myocyte only) and the overlay (native; green and donor; red).

3. For Figure 3A, please show the conducting statistics.

A table has been inserted into this figure with the time specific values and standard deviations for both the OCR and ECAR data.

Reviewer #2: The well written and coherent work describes research on mitochondrial transplantation using non-neoplastic, human myocyte-derived mitochondria into two mesenchymal subtype glioblastoma cell lines. The study investigates the impact on metabolic architecture and response to ionizing radiation, combining functional bioenergetic assays and transcriptomic profiling. The work is well presented, conceptualized and scientifically sound. I have several points that could help to improve the already great study.

Questions:

Can the authors comment on the purity and integrity of the isolated mitochondria after transplantation?

The mitochondrial isolation was performed using a commercially available kit with manufacturer provided technical references. Although we performed the JC-1 assay on the transplants under stress and non-stress conditions, no additional test of mitochondrial function or membrane integrity was performed. We can assume that the response in the U3035 cells indicates that functional mitochondria and the U3046 cells received the same complement of donor mitochondria from the same extractions. We have mentioned this limitation to the process in our revised discussion:

Discussion Page 15, Line 23:

“Our work also indicates that transplanted mitochondria maintain polarization even under nutrient-limited or hypoxic conditions and reduced radiation-induced depolarization, suggesting a stabilizing effect on mitochondrial integrity under metabolic stress.”

A possible limitation is the short post-translation effect (48 hours), could long-term studies improve the understanding of therapeutic durability?

In our previous work on this topic, we employed the strategy of mitochondrial transplantation from one distinct glioblastoma cell line to another, using the same donor: recipient dye combinations (Marshall et al, 2025). In that study, we assayed for the retention of donor mtDNA in recipient cells up to 14 days post-transplantation. Within the revised discussion, we review the potential applications for mitochondrial transplantation largely encompassing adjuvant therapy to the standard of care given in a relatively short interval. We have mentioned the need for “long-term” studies in the discussion and how these would benefit translational potential:

Discussion Page 17 Line 14:

“Rather than a direct cytotoxic strategy, mitochondrial augmentation may represent a precision therapeutic approach to unmask latent metabolic liabilities in adaptable GBM subtypes. Stratification based on baseline metabolic engagement and radiation transcriptional profiles may help identify patients most likely to benefit from metabolism- and redox-directed combination strategies. This builds on our prior work that significant heterogeneity in response exists among glioblastoma cell lines, even those that share mesenchymal gene expression profiles [24]. Further in vivo studies will aid in elucidating the impact that mitochondrial transplantation could have in optimizing standard of care treatment for glioblastoma as a therapeutic adjunct.”

In addition, the observation that U3046 cells show higher mitochondrial uptake but less functional augmentation raises questions about mitochondrial turnover, integration, or degradation in different subtypes.

Thank you for addressing this phenomenon. One of our central conclusions to this study is the potential variable response to mitochondrial transplantation based on the recipient cell line. This underscores the heterogeneity of glioblastoma that becomes apparent when standard treatments, such as radiotherapy, are undertaken. We have discussed this in one of our other previous studies (Cifarelli et al, 2021) and have incorporated it into the revision:

Discussion: Page 15, Line 22:

“Notably, U3046 cells exhibited more persistent mitochondrial signal post-transplantation, possibly reflecting enhanced uptake or reduced turnover. Our work also indicates that transplanted mitochondria maintain polarization even under nutrient-limited or hypoxic conditions and reduced radiation-induced depolarization, suggesting a stabilizing effect on mitochondrial integrity under metabolic stress.”

Discussion: Page 17, Line 16:

“Stratification based on baseline metabolic engagement and radiation transcriptional profiles may help identify patients most likely to benefit from metabolism- and redox-directed combination strategies. This builds on our prior work that significant heterogeneity in response exists among glioblastoma cell lines, even those that share mesenchymal gene expression profiles [24].”

There is minimal discussion on whether mitochondrial transplantation could inadvertently support tumor growth in some contexts or induce off-target effects.

This was an initial concern of ours as well and emphasizes the need for further in vivo studies. We have discussed this in the limitations section:

Discussion Page 17 Line 3:

“While promising, this study has important limitations. Chief among them is the absence of in vivo outcome data and the lack of assessment regarding how mitochondrial transplantation may influence the long-term proliferative capacity of GBM cells. Addressing the former would require rodent model testing, which is complicated by the challenge of selecting a representative cell line—especially given the significant functional differences we observed between just two human GBM lines. Moreover, the increasing recognition of immune system involvement in GBM response would necessitate either the use of rodent glioma lines or humanized models, both of which may limit translatability. Regarding the potential impact on proliferation, our prior study using neoplastic donor and recipient cells did not demonstrate any increase in proliferative index [7]. However, future investigations—particularly those involving different donor:recipient combinations—will need to closely evaluate this possibility.”

The use of the cell-penetrating peptide Pep-1 is described, but the efficiency, potential toxicity, and specificity of this delivery mechanism are not deeply discussed.

CPPs, including Pep-, have been used in clinical models. We have updated the discussion with additional references:

Discussion Page 15, Line 18:

“Using a Pep-1 peptide–mediated delivery platform, we achieved efficient mitochondrial transfer from skeletal myocytes, confirmed by fluorescent signal tracking. CPPs, such as Pep-1 have a substantial amount of data supporting the efficiency of transport into a donor cell [18, 19]. Moreover, recent studies have confirmed that this approach can be clinically translatable with no evidence of toxicity[20].”

Clarification on statistical methods and whether all replicates are biological or technical would clarify the validity of the findings.

These notes are made within the figure captions for each data figure where applicable.

The authors should address how the findings of the work could translate into in vivo models

Discussion Page 17, Line 14:

“Rather than a direct cytotoxic strategy, mitochondrial augmentation may represent a precision therapeutic approach to unmask latent metabolic liabilities in adaptable GBM subtypes. Stratification based on baseline metabolic engagement and radiation transcriptional profiles may help identify patients most likely to benefit from metabolism- and redox-directed combination strategies. This builds on our prior work that significant heterogeneity in response exists among glioblastoma cell lines, even those that share mesenchymal gene expression profiles [24]. Further in vivo studies will aid in elucidating the impact that mitochondrial transplantation could have in optimizing standard of care treatment for glioblastoma as a therapeutic adjunct.”

Minor comments:

In methods, the composition of mitochondrial isolation buffer should be given

The mitochondrial isolation is performed using a commercially-available kit (BioVision; K288-50), now Abcam; ab288084). The manufacturer does not provide the composition of this proprietary buffer.

Beginning of discussion: “granualr" is mistyped

Corrected

The picture depicted in fig 6a was created de novo by authors, or adopted from different work? If the latter is true, it should be referenced. The same applies for 6b.

Thank you for identifying this oversight on our behalf. The figures from the KEGG pathways have been granted copyright release from the appropriate owners and documentation

---

## [Decision Letter · Decision Letter 1]

20 Sep 2025

Dear Dr. Cifarelli,

Thank you for submitting your manuscript to PLOS ONE. After careful consideration, we feel that it has merit but does not fully meet PLOS ONE’s publication criteria as it currently stands. Therefore, we invite you to submit a revised version of the manuscript that addresses the points raised during the review process.

Please respond to reviewer's comments.

We look forward to receiving your revised manuscript.

Kind regards,

Xiaosheng Tan

Academic Editor

PLOS ONE

Journal Requirements:

Reviewers' comments:

Reviewer's Responses to Questions

**Comments to the Author**

Reviewer #1: All comments have been addressed

Reviewer #2: All comments have been addressed

2. Is the manuscript technically sound, and do the data support the conclusions?

Reviewer #1: Yes

Reviewer #2: Yes

3. Has the statistical analysis been performed appropriately and rigorously?

Reviewer #1: No

Reviewer #2: Yes

4. Have the authors made all data underlying the findings in their manuscript fully available?

Reviewer #1: Yes

Reviewer #2: Yes

5. Is the manuscript presented in an intelligible fashion and written in standard English?

Reviewer #1: Yes

Reviewer #2: Yes

Reviewer #1: 1. The quality of Figure 5 is not good. It is hard to identify the letter. Please improve it.

2. The quality of Figure 4A, 4B are not good. The letter of the top is hard to identify.

3. Please reorganize the order of all figures.

Reviewer #2: The authors answered all my questions and addressed the concerns. I recommend the work for publication and congratulate them on the great work.

**Do you want your identity to be public for this peer review?** For information about this choice, including consent withdrawal, please see our Privacy Policy

Reviewer #1: No

Reviewer #2: No

---

## [Author Response · Author response to Decision Letter 2]

23 Sep 2025

Responses to Reviewers (PONE-D-25-41544R1)

Reviewer #1:

1. The quality of Figure 5 is not good. It is hard to identify the letter. Please improve it.

Figure 5 has been re-uploaded at 1200dpi, which exceeds the requirements of the Journal.

2. The quality of Figure 4A, 4B are not good. The letter of the top is hard to identify.

Figures 4A and 4B have been re-uploaded at 1200dpi, which exceeds the requirements of the Journal.

3. Please reorganize the order of all figures.

The order has been re-checked and is in accordance with the text.

Reviewer #2: The authors answered all my questions and addressed the concerns. I recommend the work for publication and congratulate them on the great work.

Thank you for your support.

---

## [Decision Letter · Decision Letter 2]

26 Sep 2025

Dear Dr. Cifarelli,

Thank you for submitting your manuscript to PLOS ONE. After careful consideration, we feel that it has merit but does not fully meet PLOS ONE’s publication criteria as it currently stands. Therefore, we invite you to submit a revised version of the manuscript that addresses the points raised during the review process.

We look forward to receiving your revised manuscript.

Kind regards,

Xiaosheng Tan

Academic Editor

PLOS ONE

Journal Requirements:

**Additional Editor Comments:**

Please respond to reviewer's comment.

Reviewers' comments:

Reviewer's Responses to Questions

**Comments to the Author**

Reviewer #1: All comments have been addressed

2. Is the manuscript technically sound, and do the data support the conclusions?

Reviewer #1: Yes

3. Has the statistical analysis been performed appropriately and rigorously?

Reviewer #1: Yes

4. Have the authors made all data underlying the findings in their manuscript fully available?

Reviewer #1: Yes

5. Is the manuscript presented in an intelligible fashion and written in standard English?

Reviewer #1: Yes

Reviewer #1: It looks fine now. I agree to publish it. But the quality of Figure 5 is not good. It is better to improve the resolution of it.

**Do you want your identity to be public for this peer review?** For information about this choice, including consent withdrawal, please see our Privacy Policy

Reviewer #1: **Yes: ** Chunyu WEI

---

## [Author Response · Author response to Decision Letter 3]

29 Sep 2025

Responses to Reviewer (PONE-D-25-41544R2)

Reviewer #1: Reviewer #1: It looks fine now. I agree to publish it. But the quality of Figure 5 is not good. It is better to improve the resolution of it.

Figure 5 has been revised again.

---

## [Editor Report · Decision Letter 3]

1 Oct 2025

Differential Bioenergetic Profile of Human Glioblastoma following Transplantation of Myocyte-derived Mitochondria

PONE-D-25-41544R3

Dear Dr. Cifarelli,

We’re pleased to inform you that your manuscript has been judged scientifically suitable for publication and will be formally accepted for publication once it meets all outstanding technical requirements.

Kind regards,

Xiaosheng Tan

Academic Editor

PLOS ONE
---

## [Editor Report · Acceptance letter]

PONE-D-25-41544R3

PLOS ONE

Dear Dr. Cifarelli,

I'm pleased to inform you that your manuscript has been deemed suitable for publication in PLOS ONE. Congratulations! Your manuscript is now being handed over to our production team.

Kind regards,

on behalf of

Dr. Xiaosheng Tan

Academic Editor

PLOS ONE